# Enhancing the accuracy and efficiency of Pacific walrus (*Odobenus rosmarus divergens*) surveys: A comparison of visual and aerial imagery-based counts at coastal haulouts

**Alexey V. Altukhov**[1]*, **Natalia V. Kryukova**[2], **Irina S. Trukhanova**[1], **Vladimir N. Burkanov**[1]

**1** North Pacific Wildlife Consulting, LLC, Seattle, WA, United States of America, **2** Laboratory of Behavior and Behavioral Ecology, Severtsov Institute of Ecology and Evolution Russian Academy of Sciences, Moscow, Russia

* aaltukhov@gmail.com

**Data Availability Statement:** Walrus Coastal Haulouts: Outlines and Herd Densities from Drone

## Abstract

This study presents a semi-automated approach utilizing unoccupied aerial vehicle (UAV) surveys to accurately estimate the abundance of Pacific walruses at large coastal haulouts in Chukotka, Russia. Seven major haulout sites were surveyed during the summers and falls of 2017–2019. Walrus counts were performed using three distinct methods: traditional visual land-based counts, complete head counts utilizing georeferenced UAV imagery, and counting walruses within model polygons within the haulout outline and employing various extrapolation techniques to predict walrus abundance across the haulout area. The results indicated that traditional visual counts neither yielded consistent results nor allowed for uncertainty estimation, unlike the site- and date-specific direct extrapolation method and the non-specific linear regression model. These latter methods consistently provided estimates, on average, within 5% of the "true" abundance determined through complete photo-based head counts. Beside yielding accurate estimates, these semi-automated methods significantly reduced counting time by at least 63%, in contrast to complete head counts. The non-specific model, which allowed the estimation of walrus abundance based on the type of the terrain and the haulout area was less accurate compared with site and date specific estimates, but provided a tool to estimate abundance when no field visits are conducted, e.g., by using high-resolution satellite imagery to measure haulout area. This model revealed that the haulouts located on flat sandy beaches exhibited mean walrus densities approximately 30.5% times higher than those on rocky shores surrounded by cliffs: 0.879 (SD = 0.1302) and 0.648 (SD = 0.1753) walrus per $m^2$ correspondingly. The estimated daily walrus abundance at major Chukotkan haulouts in 2017–2019 ranged between 15 and 94,660 (mean = 10,397, SD = 14,477) walruses with the maximum seasonal abundances reported at Cape Serdtse-Kamen as 94,960 on 10-Oct-2017, 26,850 on 10-Oct-2018, and 87,595 on 10-Oct-2019.

Imagery Dataset 2017-2019 available from BioStudies, S-BSST1263. 2023. Retrieved from https://www.ebi.ac.uk/biostudies/studies/S-BSST1263

**Funding:** Fieldwork was mutually funded by USGS (contract G17PX00579 for 2017) and US FWS (contract F16PC00020 for 2018-2019). Support for the data processing and this analysis came from the USGS Ecosystems Mission Area, through the Quick Response Program, for the U.S. Fish and Wildlife Service, Marine Mammals Management office through the North Pacific Wildlife Consulting, LLC (http://www.northpacificwildlife.com/) via contracts 140F0719P0055 and 140F0222C0001. The funders did not have any additional role in the study design, data collection and analysis, decision to publish, or preparation of the manuscript. However, A.S. Fischbach from the USGS Ecosystems Mission Area served as the technical liaison on questions of harmonizing data processing with the USGS drone survey data processing.

**Competing interests:** The authors have declared that no competing interests exist.

## Introduction

For several decades, Pacific walrus (*Odobenus rosmarus divergens*) monitoring at coastal haul-outs has been a significant source of information on its population abundance, demographic structure, and vital rates including birth rates, calf survival, and mortality [1–13]. Researchers have conducted biological and ecological studies, which serve as the foundation for current understanding of walrus behavior [14–16]. These studies have traditionally relied on visual observations of walruses aggregating on the coast during the summer and fall months, with observers carrying out walrus head counts and demographic surveys [7, 13, 17, 18].

Traditional visual methods of walrus abundance estimation, involving counting walruses by tens or hundreds, using geometric figures, or estimating based on the area occupied by the herd [1, 4, 19–21], have certain advantages, including independence from weather conditions and the relative speed of the method. However, these estimates suffer from two major draw-backs: imprecision in the counts and absence of estimable uncertainly around the point estimate. Deviations in the abundance estimates obtained through traditional methods can reach up to 30% from the true value, either underestimating or overestimating the herd size [22–25]. The accuracy of traditional counts depends on several factors, including the density of the haulout, the homogeneity of walrus distribution across the entire haulout area, the viewing angle from observation point, the accessibility of both the haulout boundaries and the entire area to the observer, visibility, and the observer's experience. The visual surveys at walrus haul-outs have had significant discrepancies when done multiple times in one day, simultaneously by more than one researcher, or when compared with data collected using instrumental approaches, e.g., with unoccupied aerial vehicles (UAVs or drones) [22, 26–28]. The absence of estimable uncertainty surrounding the walrus count renders comparisons across years or studies unfeasible. Moreover, without estimable uncertainty, reliable estimations of associated demographic parameters such as age and sex composition, reproductive rates, or mortality become impossible. The need for standardizing walrus abundance estimation methods at coastal haulouts, improving their precision and using appropriate uncertainly estimation techniques has been recognized previously [29] and UAVs have been demonstrated to become a suitable tool for that.

Walruses typically congregate in large herds on the coast, sometimes comprising hundreds of thousands of animals [30], and haulouts are highly dynamic [31] and challenging to approach closely without disturbing the animals [32–34]. When conducted appropriately, UAV surveys do not disturb the walruses [35], provide high-resolution georeferenced imagery of the entire herd, and enable reliable animal counts on the resulting orthomosaic [9, 26, 27]. Creating georeferenced orthomosaics also enables precise measurement of the area occupied by walruses on shore and estimation of walrus density in certain parts of the occupied area [9]. This information can then be extrapolated over the entire haulout, minimizing the time spent on complete head counts. Furthermore, collecting instrument-based density measurements for a variety of sites and terrains allows obtaining universal density coefficients to estimate the walrus abundance estimates without dedicated field surveys using only remote sensing data, e.g., high resolution satellite images [36].

Building on experience in conducting counts of pinnipeds at coastal haulouts [37], walrus surveys simultaneously using UAVs and land-based visual counts were carried out 2017 through 2019 [22–26] to estimate the abundance of Pacific walruses at coastal haulouts in Chu-kotka, Russia. The primary objectives were to develop a semi-automated approach to reliably estimate walrus abundance at large coastal haulouts with estimable uncertainty and to compare the precision of traditional visual counts, complete manual photo-based counts, and modeling methods of assessing herd size. We hypothesized that the semi-automated counts, in which

walruses are counted within model polygons created on georeferenced orthomosaics, and the abundance is then modeled based on haulout size and a type of terrain, would provide more reliable results than visual surveys on-site, allow for uncertainly estimation, and would be performed in a more efficient timeframe compared with complete manual photo-based counts.

## Materials and methods

### Field surveys

Pacific walrus haulout sites - Cape Schmidt, Cape Vankarem, Kolyuchin Island, Cape Serdtse-Kamen, Keniskin Bay, Cape Inchoun and Cape Kriguigun (near Lorino village) (Fig 1) - located along north and east coast of Chukotka, Russia, were surveyed in August-October 2017, August–November 2018, and September–November 2019 (Table 1). In the course of three years, we undertook 102 visual ground counts of walruses gathered on shore on 3 sites and performed 201 counts using aerial imagery data on 8 sites. We generated 185 georeferenced orthomosaics using aerial imagery, enabling post-survey analysis. For 139 of the orthomosaics, complete head counts were available, and for 88 we performed selective counts within model polygons. Survey dates and locations are presented in Table 1.

Walrus abundance at Kolyuchin Island and Cape Vankarem in 2017 haulouts was monitored through visual land-based counts by observers. The estimation of herd size was accomplished using the approximation on land method, a technique previously employed by G.P. Smirnov [40] and A.A. Kochnev [41] to estimate walrus abundance at large coastal haulouts. The standard approximation on land method involved counting a predetermined number of individuals (e.g., 100) and noting the area they occupied. This information was then visually extrapolated to other areas within the same haulout that exhibited a similar density of animals. In instances where the haulout exhibited an uneven distribution of animals, it was visually divided into sections with varying animal densities, and the extrapolation was conducted section by section. Small haulouts, housing up to a few hundred animals, had their herd sizes assessed by individually counting all the walruses present.

In all other sites and years, in addition to the traditional methods of estimating herd size at the haulouts, we employed unoccupied aerial vehicles equipped with onboard cameras to collect imagery of the haulout. This method has been previously employed in surveying the haulouts of other pinnipeds in the Northern Pacific (e.g., [37]), as well as walrus haulouts in Alaska [30]. We used several models of DJI manufactured drones (Phantom 4, Phantom 4 pro, Mavic 2 pro; Shenzen, China). Drone surveys were conducted during favorable weather conditions, characterized by wind speeds of up to 10 m/s and the absence of precipitation. Whenever possible, to prevent disturbance, the drones were launched and landed at least 100 m away from the walrus haulout. The launch locations were determined with consideration for wind direction, ensuring the drone was launched from the leeward side of the haulout. The UAV flew transects over the walrus haulouts at an altitude of 75 to 120 m, utilizing either autonomous flight mission mode or manual mode while capturing overlapping photographs of the terrain and walruses. Adjustments in flight altitude could be made to ensure optimal image quality across diverse environmental conditions while adhering to walrus tolerance to disturbance and flight safety considerations.

Aerial images were georeferenced by estimating coordinates onboard the UAV using the Global Navigation Satellite System (GNSS). The recorded coordinates were based on the WGS84 coordinate reference system, with elevation specified in meters relative to the WGS84 ellipsoid. However, due to the lack of survey-grade ground control points to cross-validate and calibrate the GNSS locations collected onboard the UAV, the provided locations are not accompanied by a specified location accuracy report.

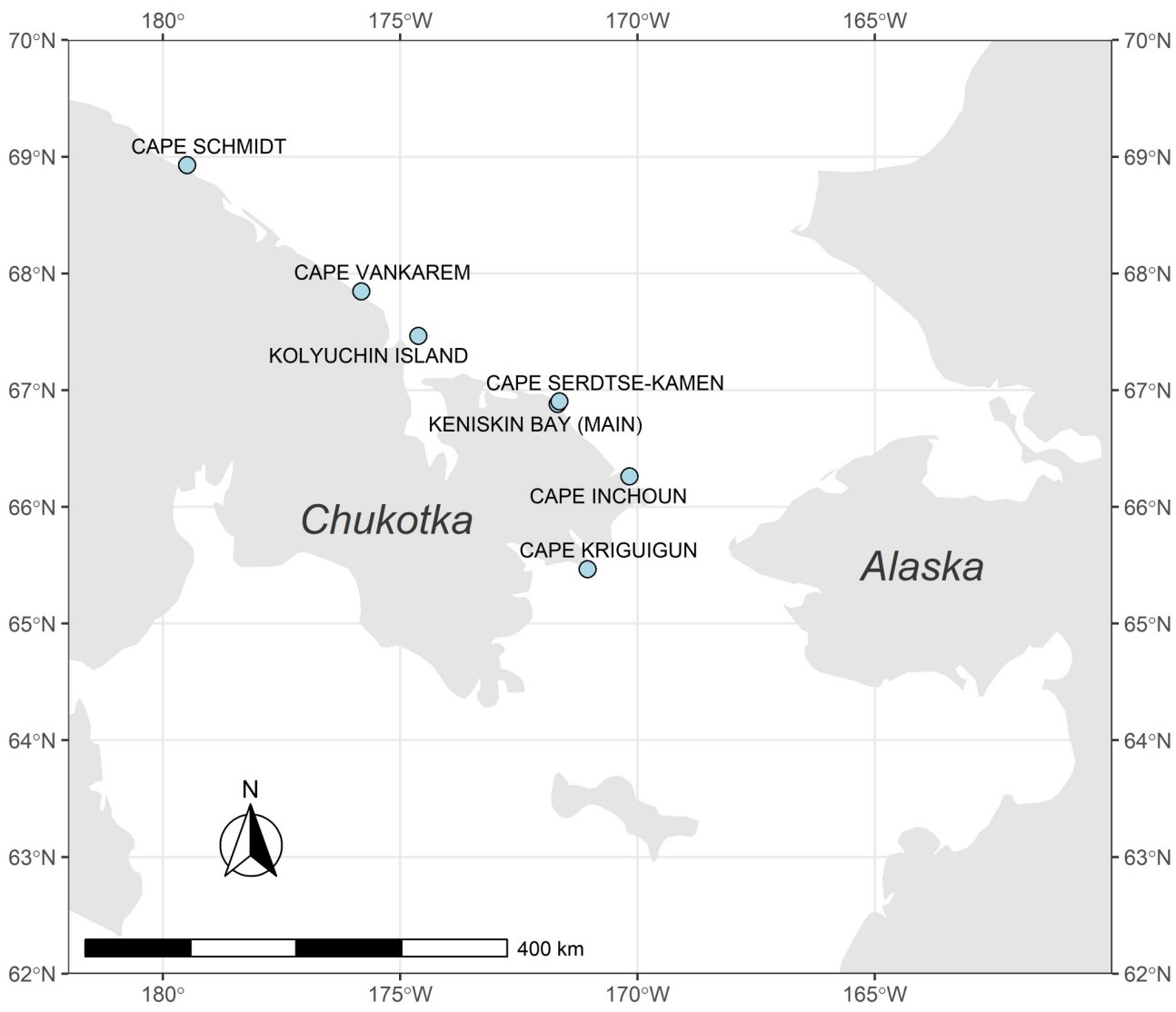

**Fig 1. Pacific walrus haulout sites surveyed in 2017–2019.** Note: Keniskin Bay (south) is not shown on the figure due to proximity to the Keniskin Bay (main). For the site detailed information and coordinates refer to the metadata. The map generated in R using several open source packages including "ggplot2" [38] using world map data from "rnaturalearth" and "rnaturalearthdata" packages [39].

All surveys were conducted as part of the resource investigation program of the Kamchatka Branch of the Pacific Geographical Institute of the Far Eastern Branch of the Russian Academy of Sciences, and were performed in a manner that minimized disturbances to the walruses resting on land.

**Table 1. Survey dates by survey method, year, and haulouts.** Empty cells indicate that a particular type of count was not performed for that location and year.

| site | site_id | year | from | to | aerial count | ground count | mosaics | model counts |
|---|---|---|---|---|---|---|---|---|
| KOLYUCHIN ISLAND | 1001 | 2017 | 08/25 | 10/09 | 26 | 46 | 14 | |
| CAPE KRIGUIGUN | 1002 | 2017 | 11/17 | 11/17 | 1 | 1 | 1 | |
| CAPE SERDTSE-KAMEN | 1007 | 2017 | 09/11 | 10/25 | 14 | | 11 | 3 |
| CAPE SERDTSE-KAMEN | 1007 | 2019 | 09/22 | 11/01 | | | 3 | 3 |
| KENISKIN BAY (MAIN) | 1003 | 2017 | 09/26 | 10/19 | 14 | | 14 | 1 |
| KENISKIN BAY (MAIN) | 1003 | 2018 | 10/15 | 11/18 | | | 8 | 8 |
| KENISKIN BAY (MAIN) | 1003 | 2019 | 09/29 | 10/23 | | | 10 | 10 |
| KENISKIN BAY (SOUTH) | 1008 | 2019 | 10/13 | 10/23 | | | 4 | 4 |
| CAPE SCHMIDT | 1004 | 2017 | 08/14 | 10/05 | 42 | | 31 | |
| CAPE INCHOUN | 1005 | 2017 | 09/14 | 10/18 | 10 | | 6 | |
| CAPE INCHOUN | 1005 | 2018 | 10/05 | 10/21 | 4 | | 4 | |
| CAPE VANKAREM | 1006 | 2017 | 08/31 | 10/24 | 55 | 55 | 27 | 7 |
| CAPE VANKAREM | 1006 | 2018 | 08/30 | 10/24 | 35 | | 52 | 52 |
| Total | | | | | 201 | 102 | 185 | 88 |

## Imagery processing

The imagery data collected by drones was processed using structure from motion software AgiSoft PhotoScan Professional version 1.8.3 [42]. During the image processing, a standard workflow within the program was employed, which involved aligning the images and performing bundle adjustment for the following parameters:

f - focal length measured in pixels (in pixels);

cx, cy - principal point coordinates, i.e., coordinates of lens optical axis interception with sensor plane (in pixels);

b1, b2 - affinity and non-orthogonality (skew) coefficients (in pixels);

k1, k2, k3, k4 - radial distortion coefficients (dimensionless);

p1, p2 - tangential distortion coefficients (dimensionless).

Subsequently, a dense cloud was generated using medium depth filtering, and a medium density height field mesh was constructed with interpolation enabled. From this mesh, orthoimages were generated in a WGS84 coordinate system by overlaying a mosaic of images onto the mesh. The resulting orthomosaic resolution ranged from 1.5 to 3.8 cm per pixel.

For specific orthomosaics, particularly at the most abundant site of Keniskin Bay, we systematically reduced spatial errors in the sparse cloud using gradual selection techniques, as described in a USGS publication [36, 43], based on the parameters outlined in the corresponding publication's metadata. Subsequently, we outlined polygons on these two sets of mosaics using consistent ground markers such as rocks, pieces of lumber, and other permanent features. By comparing the areas of these outlined polygons, we were able to quantitatively assess and compare our approach of constructing orthorectified mosaics with the approach developed by USGS, taking into account potential differences between the two methods. Overall, 12 pairs of mosaics were created and assessed with a total number of 95 pairs of polygons to compare, results of this comparison are presented in supplemental S1 Text.

We analyzed survey imagery data to count walruses, either by identifying them in the generated mosaics or directly on drone images, using specially developed software. For counting on raw drone images, NPWC, LLC designed Photo Count software that enabled us to navigate through images, mark count extents, and identify individual animals. This application also allowed us to save counts immediately in a data container (https://github.com/NPWCLLC).

To calculate the areas of walrus presence, we employed a projected coordinate reference system using the World Geodetic System 1984 (WGS 84) datum with the Universal Transverse Mercator (UTM) projection in Zone 2, North Hemisphere (EPSG:32602).

For this analysis, orthoimages obtained from 185 UAV aerial surveys with walruses present were thoroughly examined to identify the boundaries of walrus groups resting on the shore. Geospatial polygon outlines of these walrus groups, excluding those located in the surf zone or water, were digitized from the aerial imagery. The outlines encompassed isolated groups of two or more live walruses within one body length of each other.

In 2017, the drone imagery was directly processed to obtain count data before the creation of orthomosaics. As the counting effort had already been completed, instead of repeating the counts, we randomly selected portions of the haulouts that had already been counted. We outlined the counted sections of varying size and shape on the mosaic and used them as model polygons for walrus counting.

For four of the haulouts surveyed in 2018 and 2019 (77 survey days), a grid consisting roughly of 19.68 by 19.68 m cells was generated across the walrus group outlines. Each survey's grid cells were then clipped based on the corresponding walrus group outlines, retaining only the portions of grid cells within those outlines. Next, a uniform random number generator was utilized to select 30% of the clipped grid cells, which were designated as model polygons.

To digitize spatial points, AgiSoft PhotoScan Professional software was employed, focusing on the centroids of walruses that had the majority of their apparent body within the selected model polygon. If the image quality was too poor within a specific polygon to visually interpret individual walruses, that polygon was excluded from the analysis.

## Data analysis

The haulout sites were classified into two categories based on the type of terrain used by walruses: rocky_type, which consisted of rocky shores with stones and high cliffs, representing spatially restricted haulouts with limited available space, and sandy_type, which included sandy shores without cliffs, representing spatially non-restricted haulouts where animals could spread out beyond the observed herd's occupied areas. To assess whether the terrain type affected walrus density, we used the Wilcoxon rank sum exact test to compare walrus densities within model polygons (walrus number / polygon area in m$^2$) across different study sites.

For the haulouts where both visual and photo-based counts were available we compared the resulting densities using the Wilcoxon signed rank test.

We had multiple observers count walruses on model polygons. For model polygons that were processed by more than one observer, the mean counts, standard deviation (SD), and coefficients of variation (CVs) of counts were estimated. The CV, expressed as a percentage, provided a measure of relative variability in relation to the mean count. We estimated walrus abundance resting on shore during each survey as the product of the area occupied by the walrus herds resting on shore and the mean density of walruses across the sampled polygons, weighted by polygon area.

First, we wanted to check how well different methods were able to estimate the walrus abundance for a specific site and survey date. For the sites with sample polygon counts we estimated mean walrus density and associated SD per site per date and extrapolated the resulting value over the entire area occupied by the herd on that date in that particular location. We constructed the confidence interval (CI) around the estimated abundances by simulating random samples from a normal distribution with the mean equal to the mean density per site per date and SD equal to the corresponding SD, calculating quantiles of the simulated distribution corresponding to the lower 2.5% and upper 97.5%, and then multiplying those quantiles by a

given site area value. Next, we re-estimated the same abundances using a simple linear regression model fitted to the sample polygon counts modeled as a function of polygon area. The CI around the model predictions were estimated by adding/subtracting the product of the standard error of the prediction and the critical value (z-score) to/from the point estimate. The resulting site and date specific estimates were then compared with the complete walrus counts whenever those were available. The difference was expressed as a ratio of predicted count to a complete count.

Second, we built a non-specific model for all sites and dates combined to be able to estimate the walrus abundance based only on the known haulout area and type of the terrain. We employed a negative binomial generalized linear model (GLM) framework to model individual model polygon counts pooled from across all sites and survey dates as a function of the haulout site type. The variable "sample_area" was included as an offset in the model, indicating the logarithm of the model polygon size associated with each observation. The model was fitted using the MASS package [44] in R, with an initial theta value equal to 0.1. To estimate the CI of the predicted counts, we applied the delta method as follows. First, we used the fitted model to obtain the log-transformed predictions of counts log(count) for entire haulout sites with known areas. Then, we computed the variance of log-transformed response as $var(log(count)) = log(1+var(count)/count^2)$. Using the delta method, we then estimated the CIs around each point estimate of count as $exp(log(count) -/+ 2*SD(count))$.

To assess model predictive capabilities, the resulting estimated walrus abundances were compared with the mean complete photo-based counts for each site where complete counts were available.

The data generated in this study can be accessed at the online repository (see [45]). Data processing and analysis were performed using R [46] and various packages (dplyr [47], tidyr [48], sf [49, 50], sp [51, 52], MASS [44], xml2 [53]).

## Results

For 41 surveys conducted at Kolyuchin Island and Cape Vankarem in 2017 we measured the area occupied by walrus herds, obtained complete counts on aerial-based drone images, and carried out walrus counts on the ground using a classic visual land-based count approach. The pairwise comparisons of densities obtained using classic visual land-based methods revealed differences among sites (S1 Table) as well as intra-site differences between densities estimated visually and the ones obtained using drone-based counts. There were no statistically significant inter-sites differences in densities obtained using drone-based counts (Fig 2). Land-based method used at Kolyuchin Island produced an overcount of walrus numbers (S1 Fig). The method used at Cape Vankarem consistently resulted in lower densities across surveys leading to a significant undercount, but values had a tendency to improve with an increasing number of walrus on shore (S1 Fig). The median difference between the land-based approximation method used in visual counts and the drone-based counts at Cape Vankarem is estimated to range between 0.251 and 0.404 (95% CI), with a central estimate of 0.332 walruses per m$^2$. Based on complete aerial photo counts, median walrus density on a haulout was 0.823 walruses per m$^2$ (range 0.58–1.12), and based on visual land count– 0.504 (range 0.247–0.738) walruses per m$^2$ (Wilcoxon signed rank test with continuity correction, V = 351, p-value < 0.001). The Wilcoxon signed-rank test with continuity correction (V = 21, p-value = 0.093) suggests that there were no significant differences between the two counting methods, possibly due to the high variability in land-based counts.

We analyzed the agreement between the observers who performed counts within the same model polygons. A total of 699 sample polygons were counted by more than one observer. The

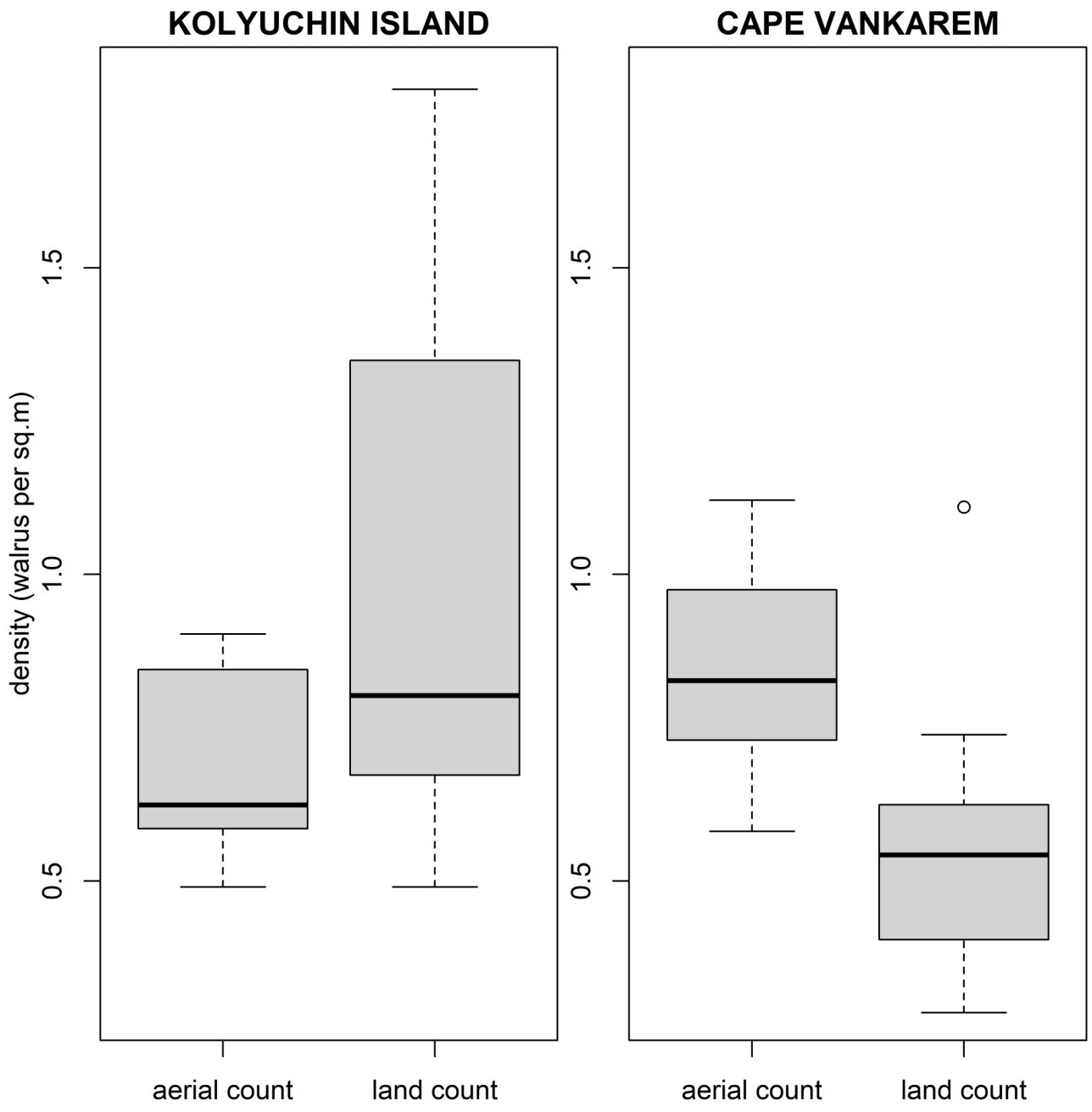

**Fig 2. Comparing densities calculated using land-based and aerial drone-based counts.**

minimum observed CV value was 0.0, indicating sites with no variability or perfect agreement among the observers. The mean CV was found to be 5.8% (SD = 8.9%), which reflects a generally good level of relative agreement in counts among the observers. However, the maximum CV observed was 101.0%, indicating substantial variability or disagreement among the observers in a few small polygons (see S2 Fig). In 853 cases, only a single observer counted a particular model polygon, so neither the SD nor the CV could be estimated for those instances.

Density variability was observed among all sites and ranged from 0.32 to 1.26 walrus per m$^2$ with the mean density of 0.76 (SD = 0.20) walrus per m$^2$. The highest density among all sites was recorded in Keniskin Bay, in both main (north) and south beaches (see Fig 3).

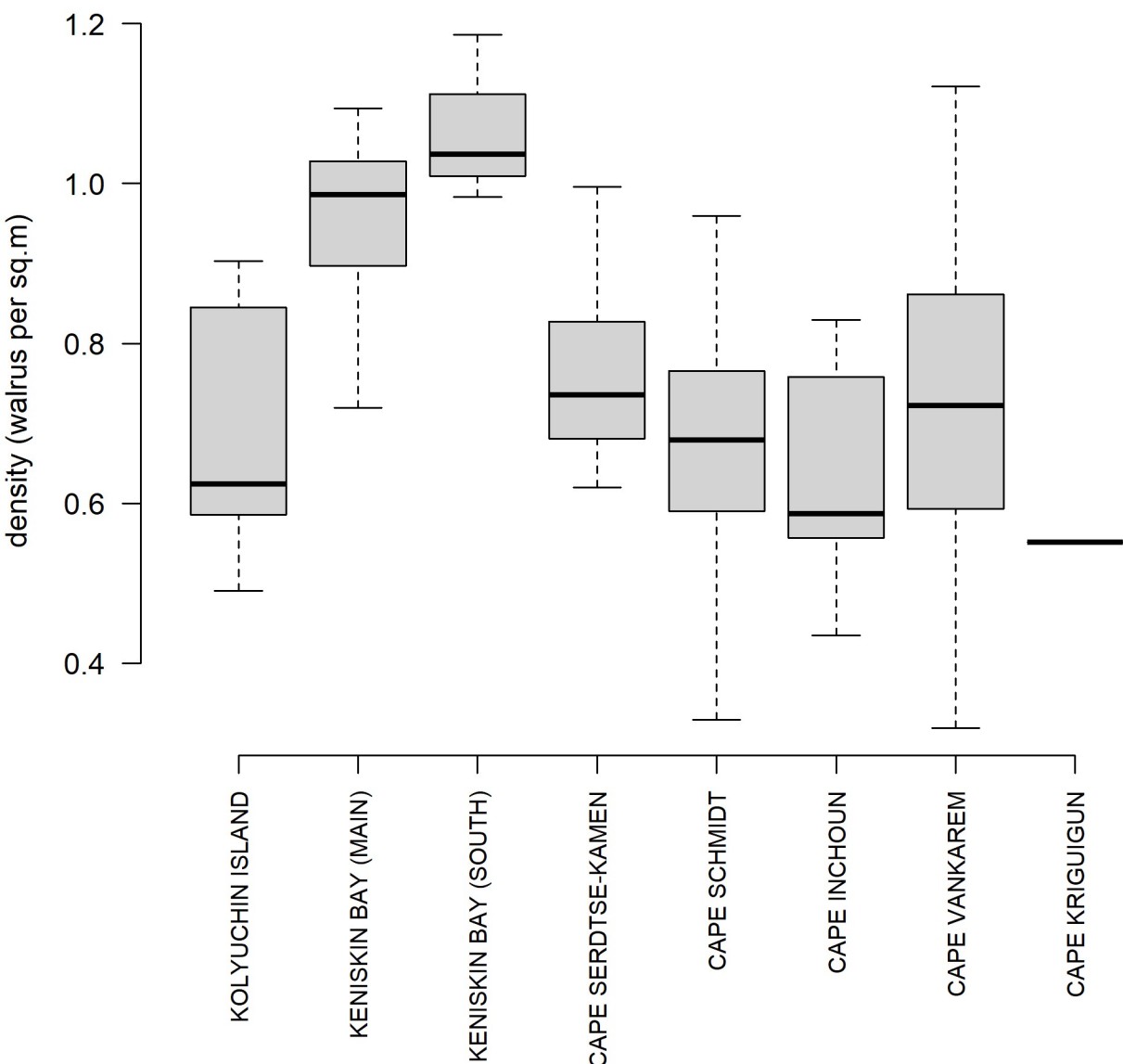

**Fig 3. The distribution of walrus densities (in walruses per m$^2$) across surveyed sites.** The data used for this plot include model polygon counts, complete haulout counts, or a combination of both methods. Each box represents the interquartile range (IQR) of walrus densities at each site, while the horizontal line within each box indicates the median density.

We observed high variability in densities within surveyed sites both within and between years (see figure in S3 Fig). At Cape Vankarem, density tended to be low early in the season and demonstrated a several-fold increase by the end of the season (see S3 Fig).

On average, sample polygons covered 35% of each haulout site area. For each model polygon, we calculated the walrus density and estimated the total expected walrus abundance for a given haulout site and date. In 41 cases, the estimated abundance could be compared with the complete walrus counts. On average, the walrus count estimated based on sample polygon densities using the extrapolation method deviated from the complete count value by -0.01% (95% CI -15.4% / 21.8%). A Summary of extrapolation results with comparison to known complete counts is provided in S1 Table. Linear model predictions deviated from the "true" counts on average by -0.06% (95% CI -27.7% / 17%). The complete comparison results along with original data are stored in a .csv table available at the data repository [45].

Based on observed counts we expected that the walrus density might vary depending on type of terrain - rocky bouldery shore (most of the surveyed sites) or a flat sandy beach (Keniskin Bay). The observed counts transformed to densities are shown in S3 Fig.

We modeled the walrus count data using a negative binomial GLM which contained the type of a haulout and log(sample_area) as an offset to account for various model polygon sizes. We refrained from incorporating additional terms into the model, such as the year or month of observation, as most sites were surveyed within a single month or year. Consequently, comparisons between different years or months were not feasible.

In the model specification count $\sim$ type + offset(log(sample_area)) the intercept was estimated to be -0.434. The variable type, specifically the level "sandy_type," had a coefficient of 0.305 (SD = 0.0195, z = 15.63), suggesting a positive effect on the count compared to the reference level, see S3 Table. Mean walrus density on sandy shores was expected to be 0.879 (SD = 0.1302) walrus per m$^2$, and on rocky terrain - 0.648 (SD = 0.1753) walrus per m$^2$.

The dispersion parameter (theta) estimated for the negative binomial distribution was 8.979, with a standard error of 0.331 thus indicating a higher extent of overdispersion in the data compared with a Poisson distribution, and confirming the correct choice of the negative binomial model. Assumptions of the negative binomial GLM, including linearity and independence, were met and confirmed by diagnostic plots (S4 Fig). However, our basic model did not account for all the variation in the observed data; only 13.68% of the deviance in the response variable (count) was attributed to the type of terrain.

When checked against the complete counts (n = 137), the predicted estimates were on average 1.7% higher that the values considered as the "true" abundance and the median estimates were 5.2% lower with 95% of predicted counts being within the 41–96% of "true" values (see S4 Fig). The predicted walrus abundances for the UAV surveyed haulouts with the corresponding CIs are presented in Fig 4.

For the sites and days where only land-based counts were performed we present the land-based estimates (S5 Fig). For the UAV flights where the orthomosaic creation was not possible, we provide complete head counts obtained from raw imagery (S6 Table). The estimated maximum walrus abundance on major Chukotkan haulouts in 2017–2019, obtained by a combination of methods, is shown in Table 2. The estimated daily walrus abundance at major Chukotkan haulouts in 2017–2019 ranged between 15 and 94,661 (mean = 10,397, SD = 14,477) walruses with the maximum seasonal abundances reported at Keniskin Bay as 94,960 on 2017-Oct-18, 26,850 on 2018-Oct-15, and 87,595 on 2019-Oct-23.

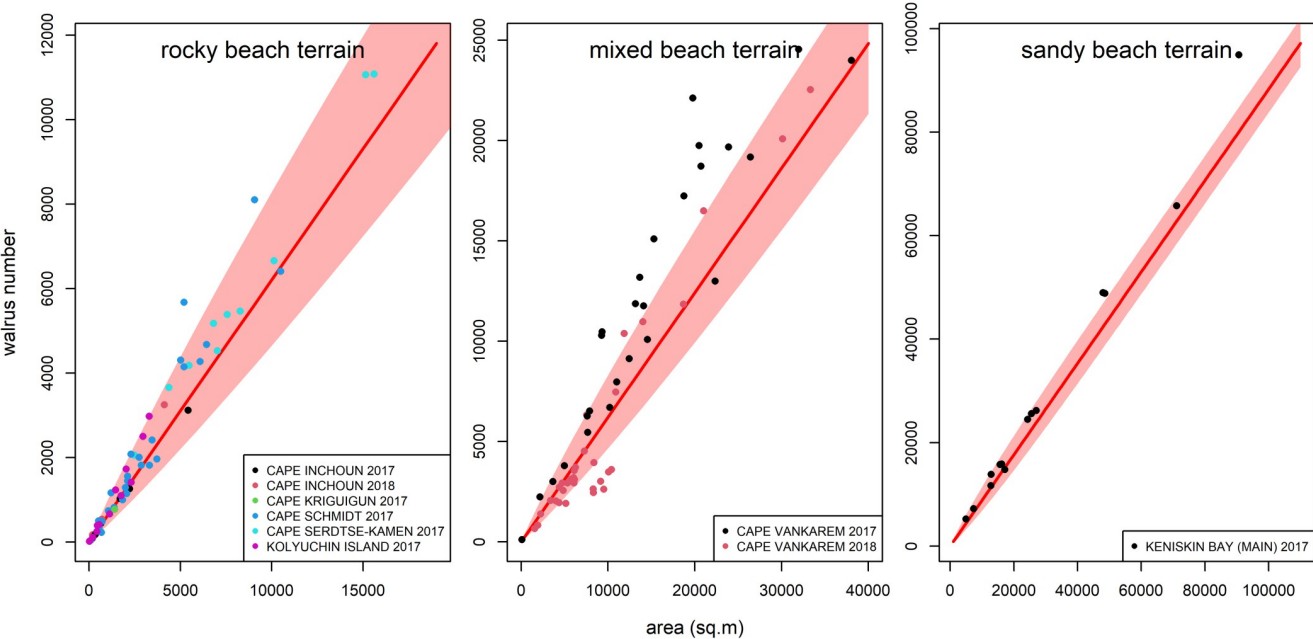

**Fig 4. Predicted herd size plotted against observed count values.** X-axis - area occupied by walruses, y-axis - number of walruses. Predictions for Cape Vankarem are shown separately from the two site types ("mixed beach terrain") for visualization purposes.

## Discussion

Pacific walruses overwinter in the Bering Sea and then migrate north through the Bering Strait to the Chukchi Sea to forage in the Chukchi Sea continental shelf waters during the summer and fall [14, 54, 55]. When foraging they can spend around between 9 and 28% of their time hauling out on shore or ice [56] forming large aggregations which, by the end of the season, in some

**Table 2. Predicted maximum total abundance of Pacific walrus by haulout site and year.** Predictions with lower and upper CI bound are obtained from non-specific negative binomial GLM counts–complete head count data obtained from orthomosaics; extrapolation–model polygon-based extrapolation results.

| site | year | prediction | | | count | extrapolation |
|---|---|---|---|---|---|---|
| | | count | lower | upper | | |
| CAPE INCHOUN | 2017 | 3513 | 2749 | 4489 | 3121 | – |
| CAPE INCHOUN | 2018 | 2675 | 2048 | 3496 | 3252 | – |
| KENISKIN BAY (MAIN) | 2017 | 79687 | 77757 | 81666 | 94961 | 92474 |
| KENISKIN BAY (MAIN) | 2018 | 27337 | 25534 | 29267 | NA | 26851 |
| KENISKIN BAY (MAIN) | 2019 | 84568 | 82634 | 86546 | NA | 87596 |
| KENISKIN BAY (SOUTH) | 2019 | 12838 | 11312 | 14569 | NA | 14361 |
| KOLYUCHIN ISLAND | 2017 | 2137 | 1607 | 2841 | 2976 | – |
| CAPE KRIGUIGUN | 2017 | 909 | 644 | 1282 | 773 | – |
| CAPE SERDTSE-KAMEN | 2017 | 10114 | 8718 | 11734 | 11084 | 8611 |
| CAPE SERDTSE-KAMEN | 2019 | 5208 | 4219 | 6430 | – | 6644 |
| CAPE SCHMIDT | 2017 | 6803 | 5649 | 8194 | 6409 | –– |
| CAPE VANKAREM | 2017 | 24660 | 22883 | 26576 | 23997 | 17053 |
| CAPE VANKAREM | 2018 | 21597 | 19858 | 23488 | 22532 | 23798 |

places amount tens of thousands of walruses [7, 9, 30, 56]. In the fall, a significant portion of the population becomes accessible for monitoring at these major coastal haulouts at Point Lay, Cape Lisburne (Alaska, USA), Cape Serdtse-Kamen, Cape Vankarem, and several others [30].

Complete walrus head counts on UAV-based imagery were considered under the present study as a more accurate method providing highly precise estimates of walrus abundance at a haulout as opposed to traditional land-based counts. Reliability of this method has been confirmed by multiple observers processing the same imagery. However, for large herds counting many thousands of animals, complete photo-based counts are very time consuming and can take many days to complete.

In this study, we sought to develop a semi-automated approach using UAV surveys to reliably estimate the abundance of Pacific walruses at large coastal haulouts in Chukotka, Russia. This approach, which involved counting walruses within model polygons created on georeferenced orthomosaics, provided more accurate results compared with visual counts conducted on-site, allowed estimating uncertainty around each abundance estimate, and could be completed in a more efficient timeframe than counting the entire herd consisting of many thousands of walruses present at a haulout. Verifiable abundance estimates with an associated measure of uncertainty make haulout survey results comparable across years, allow estimating trends in haulout occupancy, herd density, and demographic parameters of the hauled out walruses. In light of the rapid environmental changes occurring in the Arctic, having a reliable and cost-effective mechanism for counting walruses at haulouts and detecting negative trends early on is paramount for walrus population monitoring and timely management actions. Semi-automated surveys may become a valuable tool for this purpose. Our field surveys conducted over three years at multiple haulout locations in Chukotka provided valuable insights into the effectiveness of different survey methods. We encountered significant discrepancies when comparing the results of visual land-based counts with photo-based counts obtained from UAV surveys. We found that the land-based counts were prone to either significant overestimation or underestimation of herd sizes depending on the underlying terrain, availability of walruses for visual observations, and likely particular observers. For instance, on Kolyuchin Island, where the observers' view was obstructed by the rocks and part of the haulout was not visible, the counts were constantly yielding overestimated herd sizes. In contrast, counts at Cape Vankarem performed from an observation point having an almost unobstructed view over the haulout, were consistently biased low. This is consistent with earlier observations reported for a portion of this dataset by Altukhov et al. [26] who found that the difference between visual observations and photo-based counts could be as large as 30%. In contrast to visual land-based counts, counts performed by different observers using aerial imagery were consistent. The level of agreement among observers was good, with a mean CV of 5.8% thus allowing us to average the counts obtained by multiple observers. The discrepancy in counts was relatively high only for small sized polygons and primarily attributed to walruses located at the edges of polygons for which observers could arbitrarily decide to include or exclude them from the count based on their body location relative to the polygon boundary. Such discrepancies were relatively rare and did not affect the precision of counts much when working with full-size model polygons. Therefore, using photo-based counts helped to reduce the observer associated error compared with land-based counts on site and subsequently improve the accuracy of count results. Both direct extrapolation and linear regression approaches applied to site and date specific model polygon count data performed on average very well when compared with complete counts, however the discrepancy range was still quite large in some cases mainly due to high within site variability of density in certain surveys. Although clearly more precise than visual land-based counts and less time consuming than complete photo-based counts, these methods still require UAV surveys being flown on site to collect the

imagery essential for the abundance estimation. By developing a non-specific model facilitating the estimation of walrus abundance knowing only the area occupied by the herd and the type of the terrain the walruses haul out on, we provide a tool to obtain data on walrus abundance at coastal haulouts using, for example, high resolution satellite imagery where haulout area can be outlined and measured. The negative binomial GLM that we used yielded less precise estimates compared with site-specific ones but yet appears useful when direct local observations are unavailable. According to negative binomial GLM results, mean walrus density at the haulouts located on flat sandy beaches was 0.305 times higher compared with rocky shores surrounded by cliffs. The mean density predicted by the model was 0.648 for rocky terrain and 0.879 for sandy beaches. This is similar to the ranges reported by Fischbach et al. [9] for the walrus haulout at Point Lay - posterior means for the mean density of a herd ranged throughout the season from 0.67 to 1.47. The rocky beaches were characterized by the higher variability in counts and wider CI compared with sandy beaches. In the model outcome Cape Vankarem predicted abundances stood out: when compared with the complete counts they exhibited the highest count variability with a high degree of overcount at smaller sights and undercount at larger sites. The obvious reason for that was that this haulout included both types of terrain which could not be properly modeled by the selected modeling framework. It is important to note that our negative binomial model explained approximately 14% of the deviance in the observed data, indicating that there is still substantial unexplained variation in counts. Other factors not included in our model, such as environmental variables or individual behavior, may contribute to the remaining variability.Our results indicated that traditional visual counts deviated from the actual walrus abundances. In contrast, the site and date specific model polygon counts using UAV-based imagery provided very precise estimates obtained both by direct extrapolation method or using linear regression model while significantly reducing processing time by at least 63% on average.

As an example, counting all the animals at a 10,000 walrus haulout takes approximately 2.7 hours, with one walrus being counted per second. However, a semi-automated approach reduces this processing time to just 55 minutes. The non-specific modeling approach provided estimates within 5.2% of the "true" abundance determined through complete head counts but can be used going forward in situations where only remote sensing data are available.

## Conclusions

In conclusion, our study demonstrates the efficacy of using UAV surveys and semi-automated approaches to estimate the abundance of Pacific walruses at large coastal haulouts. The georeferenced orthomosaics generated from UAV imagery provided high-resolution data, enabling accurate walrus counts with estimable uncertainty and precise measurement of the area occupied by walruses on shore. The comparison between visual counts and photo-based counts revealed significant discrepancies, with the latter method consistently yielding biased estimates of walrus densities. Complete head counts using UAV-based imagery provides the objectively most precise results but are more time consuming. The model polygons and density measurements derived from UAV surveys allowed for reliable estimates of walrus abundance, with only insignificant deviation from complete counts while being significantly less time consuming. The non-specific negative binomial GLM provides a tool to obtain walrus abundance estimates at coastal haulouts based only on known terrain type and haulout area and therefore will be useful when no field surveys are carried out and only remote sensing data are available. Overall, our findings highlight the value of UAV surveys and semi-automated approaches of improving the efficiency and accuracy of population monitoring efforts for Pacific walruses and potentially other pinnipeds.

## Supporting information

**S1 Table. Pairwise comparison of land-based and aerial counts.**
(DOCX)

**S2 Table. The best model summary.**
(DOCX)

**S3 Table. Extrapolated walrus counts with differences calculated as a ratio of extrapolation to complete counts.** Negative values of ratio represent cases when extrapolated value is higher than complete manual count.
(DOCX)

**S1 Fig. Relationship between an average density and overall count in land-based count.**
(TIF)

**S2 Fig. Relationship between coefficient of variation and mean count value of outlined section of haulout.**
(TIF)

**S3 Fig. Changes in densities within and between years on each surveyed site.** x-axis is a day of the year, y-axis is density in walrus per m$^2$.
(TIF)

**S4 Fig. Histogram of differences between predicted count and direct visual aerial photo count values.**
(TIF)

**S5 Fig. Ratio of areas on mosaics without gradual selection to areas on mosaics with gradual selection applied.**
(TIF)

**S1 Text. Results of comparing mosaics created with and without sparse point cloud gradual selection approach.**
(DOCX)

## Acknowledgments

The observations and initial data processing were carried out by the following individuals: A. A. Kochnev, M.S. Kozlov and I.A. Usatov for Cape Schmidt, D.O. Skorobogatov for Cape Vankarem, A.A. Pereverzev and A.I. Shevelev for Kolyuchin Island, A.A. Kochnev and M.V. Chakilev for Keniskin Bay and Serdtse-Kamen Cape in 2017 and A.I. Shevelev for both seasons 2018 and 2019, L.E. Skurikhin, I.L. Krupin for Cape Inchoun, and S.V. Lobovikov and L.E. Skurikhin for Cape Kriguigun (Lorino village). Software development: D.N. Gaev. We are grateful Dr. Evangeline Corcoran, and the anonymous reviewer for their valuable comments and suggestions, which have significantly improved the quality of this paper.

## Author Contributions

**Conceptualization:** Alexey V. Altukhov, Natalia V. Kryukova, Vladimir N. Burkanov.

**Data curation:** Alexey V. Altukhov.

**Formal analysis:** Alexey V. Altukhov, Irina S. Trukhanova.

**Funding acquisition:** Irina S. Trukhanova, Vladimir N. Burkanov.

**Investigation:** Natalia V. Kryukova.

**Methodology:** Alexey V. Altukhov, Vladimir N. Burkanov.

**Project administration:** Alexey V. Altukhov, Natalia V. Kryukova, Vladimir N. Burkanov.

**Software:** Alexey V. Altukhov.

**Supervision:** Natalia V. Kryukova, Vladimir N. Burkanov.

**Visualization:** Alexey V. Altukhov.

**Writing – original draft:** Alexey V. Altukhov, Irina S. Trukhanova.

**Writing – review & editing:** Alexey V. Altukhov, Natalia V. Kryukova, Irina S. Trukhanova, Vladimir N. Burkanov.

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
