## [Decision Letter · Decision Letter 0]

6 May 2024

PONE-D-24-08990Enhancing the accuracy and efficiency of Pacific walrus (Odobenus rosmarus divergens) surveys: a comparison of visual and aerial imagery-based counts at coastal haulouts.PLOS ONE

Dear Dr. Altukhov,

Thank you for submitting your manuscript to PLOS ONE, and thank you for your patience while I identified two reviewers who were able to provide reviews of the manuscript. Both reviewers agreed that the manuscript provided significant contributions to enumerating walruses in coastal haul-outs. Nevertheless both reviewers provided editing suggestions and posed a few questions that would be helpful to clarify in a revised version of the manuscript. Therefore, after careful consideration, we invite you to submit a revised version of the manuscript that addresses the points raised during the review process.

We look forward to receiving your revised manuscript.

Kind regards,

Lee W Cooper, Ph.D.

Section Editor

PLOS ONE

Reviewers' comments:

Reviewer's Responses to Questions

**Comments to the Author**

1. Is the manuscript technically sound, and do the data support the conclusions?

Reviewer #1: Yes

Reviewer #2: Yes

2. Has the statistical analysis been performed appropriately and rigorously? 

Reviewer #1: Yes

Reviewer #2: Yes

3. Have the authors made all data underlying the findings in their manuscript fully available?

Reviewer #1: Yes

Reviewer #2: Yes

4. Is the manuscript presented in an intelligible fashion and written in standard English?

Reviewer #1: Yes

Reviewer #2: Yes

5. Review Comments to the Author

Reviewer #1: The authors present a comparison of traditional visual counts, manual counts derived from UAV imagery, and semi-automated methods of estimating walrus abundance that is methodologically sound and likely to be of high interest to both ecologists and other conservation decision-makers involved in monitoring and managing walrus species. The manuscript is well-presented overall with some minor errors in grammar and phrasing that should be edited to improve the clarity of the text. Please see specific comments in the attached document for details on suggested changes.

Other than the aforementioned minor language issues, I have only one concern with the manuscript that I believe needs addressing before I can recommend it be published: Throughout the manuscript multiple claims that the semi-automated method is more time efficient compared to complete counts from UAV imagery. This is likely true, but it is hard to understand the scale of difference in efficiency as the authors have not provided more specific results on the time taken to conduct each method. Adding this information to the results or discussion section would not only strengthen the authors conclusions that the semi-automated method is beneficial but would also help inform decision-making on when and where it would be more appropriate to the use the semi-automated or manual count from UAV imagery methods.

Reviewer #2: This is an interesting contribution about how to best enumerate walrus in late summer and fall coastal haulouts. The authors use UAV or satellite imagery as the gold standard and through the time-consuming process of counting individuals specify that as the “actual” number of walruses present. Comparisons with visual estimates by observers show that is likely less accurate, but short cuts taken by devising polygons and extrapolations can provide pretty accurate counts also. I think with some minor attention, this manuscript can be improved and made ready for publication. Some of the English, while not bad, can be improved and I provide some recommendation for specific line numbers. Also, it would be worth addressing the importance of making highly accurate counts of animals in haulouts, which was never really justified. While accurate enumeration can enable understanding the overall population of Pacific walruses, if an individual haulout is inaccurately counted by 25 or 30% (line 360), what are the consequences? This should be articulated better. It goes beyond stating that accurate and efficient counts (Line 347) are “paramount.” Please tell us why it is so important to have highly accurate counts.

My line by line recommendations for editing:

Line 25-26. The implication of this statement of course is that one approach for estimation, a traditional visual account, is not the definitive enumeration method. The critical word here is “actual” that indicates that there is a single correct number when there is a probably an error associated with each estimation, but I think the authors are correct that the best estimates are obtained by the time-consuming task of counting individual animals on a digital photographic image. It would be better to state that the enumeration techniques did not yield the same consistent results as the other two methods, not that it was inherently incorrect.

Line 27-28. Again, I would state that the results here are of consistency, not adherence to a “true” number that is correct.

Line 29. Why is it called the universal model? It might be the traditionally used model or the standard model, but universal sounds to all-encompassing.

Line 42. Not sure what is meant by vital rates.

Line 59. Using “or” between UAVs and drones implies that they are different vehicles.

Line 70-72. This sentence is awkwardly written, e.g. “allows to”

Line 81. This comes back to my comments on the abstract. The hypothesis is that semi-automated counts using a modeling approach based upon beach topography and haulout size would be more reliable than simply enumerating animals. But how do you assess whether the hypothesis can be accepted? It presumes that there is a gold standard to compare an enumeration method against, doesn’t it?

Line 89. Instead of “collected” use “undertook”

Line 113. Delete “mentioned” ---it isn’t necessary.

Line 116. Add the manufacturer’s location (Shenzen, China).

Line 138. For the reader not familiar with the software used, it would be helpful to fully spell out and define these parameters, e.g. from https://agisoft.freshdesk.com/support/solutions/articles/31000158119-what-does-camera-calibration-results-mean-in-metashape-:

f - Focal length measured in pixels (in pixels).

cx, cy - Principal point coordinates, i.e. coordinates of lens optical axis interception with sensor plane (in pixels).

b1, b2 - Affinity and non-orthogonality (skew) coefficients (in pixels).

k1, k2, k3, k4 - Radial distortion coefficients (dimensionless).

p1, p2 - Tangential distortion coefficients (dimensionless).

Line 155-157. It is not clear if this was software that the authors designed and wrote themselves, or if it was a modification of an existing product. If it was an existing product that the authors used, give the full name and source of the software. On line 157, the weblink provided does not lead to a working website.

Line 189. Insert “the” before Wilcoxon

Line 190. This short sentence is not clear. It is stating that there were multiple observers? (Observers should be plural in that case).

Line 211. Constructed would probably be a better word that built in this sentence.

Line 217. CI was used as early as Line 200, where it should be initially defined instead of here.

Line 218. Add to “we applied the delta method” “as follows.”

Line 221. Use superscript here for exponents rather than ^

Line 226. Change “using the R software” to using R

Line 233. Use among instead of between

Line 235. Add “The” to the start of the sentence. For Figure 2, the numbers on the y-axis should be labelled.

Line 238. Add the article “a” before tendency.

Line 229, 240. Elsewhere, this landmark is referred to as Cape Vankarem rather than Vankarem Cape. Cape Vankarem is probably more familiar to English speakers, but in any case, the geographical landmark should be referred to consistently throughout the paper.

Line 241, 242, 243, 258, 259, 293. Use superscript for exponents.

Line 274. Add “the” before extrapolation.

Line 275. Change this sentence to “A Summary of extrapolation results with comparison to known complete counts is provided in S1 Table.”

Line 282. Add “the” before walrus

Line 285. Not sure what more than one level means. More than one parameter?

Line 289. The equation used isn’t clear to me, particularly the tilde (~). Normally this mean “about,” but this seems to be part of the regression equation?

Line 298. Spell out generalized linear model on first use (GLM)

Line 303. Add the article “the” before “true”

Line 326. Change “count” to “amount”

Line 327. Delete “and more”

Line 330-332. This sentence is redundant with a very similar sentence at Line 40-42. Most of this paragraph presents background information that is also in the introduction.

Line 337. Change “close to perfect estimate” to “highly accurate estimates”

Line 338 Change “by using multiple observers to process” to “by multiple observers processing”

Line 340. Change “a person days to complete the task” to “many days to complete”

Line 376. Change “allowing to estimate the” to “facilitating the estimation of”

Line 410. Change “provide most precise results” to “provides the objectively most precise results”

Line 415 Insert “will be” between “therefore” and “useful”

Line 417. Change “in” to “for”

References section, for the publications that are written in Russian, state “in Russian” rather than just “Russian.” Species names should be italicized.

Line 520-523. The name of the proceedings is provided twice. Probably helpful here to provide a weblink to where the document is available (https://library.wcs.org/doi/ctl/view/mid/33065/pubid/DMX1388500000.aspx)

Figure 3. The units for density should be provided on the y-axis itself, not just in the figure caption, i.e., walruses per square meter

Figure 4. Again, as with the other figures, make sure the axes are clearly labelled as to which type of survey corresponds to each axis and use consistent numbering. Here it is inconsistent with scientific notation used on one graph and ordinal numbers on the other two.

I see the same limited axis labelling in the supplemental figures. On supplemental figure 5, difference is misspelled. On supplemental Figure 4, spell out full Histogram of diff. Axis labels needed on Figure 3 (suppl.), spell out cv on supplemental Figure 2, and on supplemental Figure 1, give the units for density (e.g. walrus per square meter).

6. PLOS authors have the option to publish the peer review history of their article (what does this mean?). If published, this will include your full peer review and any attached files.

Reviewer #1: **Yes: **Evangeline Corcoran

Reviewer #2: No

---

## [Author Response · Author response to Decision Letter 0]

19 Jun 2024

Author: The data if fully available online, and the link to the depository was provided during initial submission.

Author: The map on Figure 1 was generated R using several open source packages. The information is updated in captions.

Reviewer #1: The authors present a comparison of traditional visual counts, manual counts derived from UAV imagery, and semi-automated methods of estimating walrus abundance that is methodologically sound and likely to be of high interest to both ecologists and other conservation decision-makers involved in monitoring and managing walrus species. The manuscript is well-presented overall with some minor errors in grammar and phrasing that should be edited to improve the clarity of the text. Please see specific comments in the attached document for details on suggested changes.

Other than the aforementioned minor language issues, I have only one concern with the manuscript that I believe needs addressing before I can recommend it be published: Throughout the manuscript multiple claims that the semi-automated method is more time efficient compared to complete counts from UAV imagery. This is likely true, but it is hard to understand the scale of difference in efficiency as the authors have not provided more specific results on the time taken to conduct each method. Adding this information to the results or discussion section would not only strengthen the authors conclusions that the semi-automated method is beneficial but would also help inform decision-making on when and where it would be more appropriate to the use the semi-automated or manual count from UAV imagery methods.

Author: Time required to conduct a photo-based complete head count depends on herd size, image quality (images of higher quality are easier to process) and observer’s experience. On average, it is safe to assume that we can count (put a marker on the image) 1 walrus per second. Therefore, to process a haulout with 10,000 walruses on it, an observer will spend approximately 166 minutes. With polygon-based count and subsequent extrapolation, it takes about 1/3 of that time (i.e., 55 minutes). For comparison’s sake, for larger haulouts with ~100,000 animals, we are looking at 27.6 hours vs 9.2 h of work time. 

We thank the reviewer for the question and we have added the following language to the discussion:

“As an example, counting all the animals at a 10,000 walrus haulout takes approximately 2.7 hours, with one walrus being counted per second. However, a semi-automated approach reduces this processing time to just 55 minutes.”

Specific Comments

Line 42 – Use of ‘behavioural studies’ and ‘behaviour’ in the same sentence seems redundant, I suggest this line be revised to be more succinct.

Author: Replaced with biological studies

Line 50 – Change ‘speed of method’ to ‘speed of the method’.

Author: Done

Line 51 – It would be more concise to say the counts can differ by up to 30% compared to the true number of walruses present.

Author: We feel that the current wording is very clear and reflects the detected deviations in traditional count results from true values of walrus abundance 

Line 70 – Change ‘allows to obtain’ to ‘allows obtainment of’.

Author: Changed to allows obtaining….

Table 1 – It is unclear why some values are missing from this table. Please provide an explanation in the text introducing the table or the table caption.

Author: Added: Empty cells indicate that a particular type of count was not performed for that location and year. 

Line 121 – It would be helpful to include more information on why drones were flow at different heights and using different flight modes for certain surveys. 

Author: The optimum altitude for a drone survey depends on several parameters: 1) how load a drone is (i.e. model) – typically – the louder, the higher we fly; 2) how much background noise there is (the more noise (wind, waves, birds), the lower we can fly without animals noticing the drone; 3) the drone camera capabilities (certain drones have better cameras and therefore allow for higher flights without significant loss of image quality); and 4) terrain (in cliffy locations, the flights were performed at higher altitudes for safety reasons). Considering the large number of simultaneously surveyed sites and limited project budget (as it usually is the case.) various drone models were used for the survey. Flight parameters were adjusted for specific conditions (see 1-4 above) to ensure image quality sufficient for the study needs while taking in to account walrus disturbance tolerance. 

We provided explanation in the text:

Adjustments in flight altitude could be made to ensure optimal image quality across diverse environmental conditions while adhering to walrus tolerance to disturbance and flight safety considerations.

Table 2 – More information is needed in the table caption to explain which method the various counts and extrapolations were derived from.

Author: We provided the following clarification: Predictions with lower and upper CI bound are obtained from non-specific negative binomial GLM counts – complete head count data obtained from orthomosaics; extrapolation – model polygon-based extrapolation results.

Reviewer #2: This is an interesting contribution about how to best enumerate walrus in late summer and fall coastal haulouts. The authors use UAV or satellite imagery as the gold standard and through the time-consuming process of counting individuals specify that as the “actual” number of walruses present. Comparisons with visual estimates by observers show that is likely less accurate, but short cuts taken by devising polygons and extrapolations can provide pretty accurate counts also. I think with some minor attention, this manuscript can be improved and made ready for publication. Some of the English, while not bad, can be improved and I provide some recommendation for specific line numbers. Also, it would be worth addressing the importance of making highly accurate counts of animals in haulouts, which was never really justified. While accurate enumeration can enable understanding the overall population of Pacific walruses, if an individual haulout is inaccurately counted by 25 or 30% (line 360), what are the consequences? This should be articulated better. It goes beyond stating that accurate and efficient counts (Line 347) are “paramount.” Please tell us why it is so important to have highly accurate counts.

Author: The imprecision of visual counts is not the biggest issue, but the inability to assess how imprecise a particular count was, i.e., estimate the uncertainly around the point estimate, is a big problem for any scientific data. Visual counts not only suffer from poor precision but also lack estimates of uncertainty around the estimated number of animals. This means that the uncertainty of the count itself cannot be propagated further when using the local herd abundance to estimate population parameters, such as total population abundance, age and sex composition, reproductive rates, or mortality. Additionally, the visual count results obtained in different years are impossible to reliably compare to one another as they are highly dependent on individual observer’s error which is also not getting estimated (but shown to be rather large). This makes visual counts less valuable for population monitoring for research, conservation, or management needs. Model polygon-based counts (and complete imagery-based counts too, of course), on the contrary, allow for estimating precision around point estimates, are verifiable because allow for repeat counts by different observers, and the abundance estimation results can be compared across studies, years of sites. Overall, the biggest advantages model polygon-based counts provide, except for more precise results, are estimable uncertainty and verifiable results. 

We thank the reviewer for bringing this question up. We added some language to the Introduction and in the Discussion to articulate this better.

My line by line recommendations for editing:

Line 25-26. The implication of this statement of course is that one approach for estimation, a traditional visual account, is not the definitive enumeration method. The critical word here is “actual” that indicates that there is a single correct number when there is a probably an error associated with each estimation, but I think the authors are correct that the best estimates are obtained by the time-consuming task of counting individual animals on a digital photographic image. It would be better to state that the enumeration techniques did not yield the same consistent results as the other two methods, not that it was inherently incorrect.

Line 27-28. Again, I would state that the results here are of consistency, not adherence to a “true” number that is correct.

Author: The following wording is suggested: The results indicated that traditional visual counts neither yielded consistent results nor allowed for uncertainty estimation, unlike the site- and date-specific direct extrapolation method and the non-specific linear regression model. These latter methods consistently provided estimates, on average, within 5% of the "true" abundance determined through complete photo-based head counts. Beside yielding accurate estimates, these semi-automated methods significantly reduced counting time by at least 63%, in contrast to complete head counts. The non-specific model, which allowed the estimation of walrus abundance based on the type of the terrain and the haulout area was less accurate compared with site and date specific estimates, but provided a tool to estimate abundance when no field visits are conducted, e.g., by using high-resolution satellite imagery to measure haulout area.

Line 29. Why is it called the universal model? It might be the traditionally used model or the standard model, but universal sounds to all-encompassing.

Author: Replaced with non-specific. This is in contrast with site specific model.

Line 42. Not sure what is meant by vital rates.

Author: By vital rates here we assume information on e.g., birth rate, survival of young, or mortality. Clarified in the text.

Line 59. Using “or” between UAVs and drones implies that they are different vehicles.

Author: Corrected: with unoccupied aerial vehicles (UAVs or drones)

Line 70-72. This sentence is awkwardly written, e.g. “allows to”

Author: Corrected – allows obtaining

Line 81. This comes back to my comments on the abstract. The hypothesis is that semi-automated counts using a modeling approach based upon beach topography and haulout size would be more reliable than simply enumerating animals. But how do you assess whether the hypothesis can be accepted? It presumes that there is a gold standard to compare an enumeration method against, doesn’t it?

Author: Correct. The gold standard is the total head count performed by observers using georeferenced orthomosaics. Repeated full counts performed for selected images showed highly consistent estimates. 

Line 89. Instead of “collected” use “undertook”

Author: Done

Line 113. Delete “mentioned” ---it isn’t necessary.

Author: Deleted

Line 116. Add the manufacturer’s location (Shenzen, China).

Author: Added

Line 138. For the reader not familiar with the software used, it would be helpful to fully spell out and define these parameters, e.g. from https://agisoft.freshdesk.com/support/solutions/articles/31000158119-what-does-camera-calibration-results-mean-in-metashape-:

f - Focal length measured in pixels (in pixels).

cx, cy - Principal point coordinates, i.e. coordinates of lens optical axis interception with sensor plane (in pixels).

b1, b2 - Affinity and non-orthogonality (skew) coefficients (in pixels).

k1, k2, k3, k4 - Radial distortion coefficients (dimensionless).

p1, p2 - Tangential distortion coefficients (dimensionless).

Author: Thank you for that! Added to text.

Line 155-157. It is not clear if this was software that the authors designed and wrote themselves, or if it was a modification of an existing product. If it was an existing product that the authors used, give the full name and source of the software. On line 157, the weblink provided does not lead to a working website.

Author: The software was developed by North Pacific Wildlife Consulting specifically for this and similar projects. We clarified that and we fixed the link.

Line 189. Insert “the” before Wilcoxon

Author: Done

Line 190. This short sentence is not clear. It is stating that there were multiple observers? (Observers should be plural in that case).

Author: Fixed – observers.

Line 211. Constructed would probably be a better word that built in this sentence.

Author: Replaced.

Line 217. CI was used as early as Line 200, where it should be initially defined instead of here.

Author: Fixed

Line 218. Add to “we applied the delta method” “as follows.”

Author: Done

Line 221. Use superscript here for exponents rather than ^

Author: Done

Line 226. Change “using the R software” to using R

Author: Done

Line 233. Use among instead of between

Author: Done

Line 235. Add “The” to the start of the sentence. For Figure 2, the numbers on the y-axis should be labelled.

Author: Done and fixed.

Line 238. Add the article “a” before tendency.

Author: Added

Line 229, 240. Elsewhere, this landmark is referred to as Cape Vankarem rather than Vankarem Cape. Cape Vankarem is probably more familiar to English speakers, but in any case, the geographical landmark should be referred to consistently throughout the paper.

Author: Corrected – now using Cape Vankarem throughout, we also standardized other geographical names throughout the text and figures.

Line 241, 242, 243, 258, 259, 293. Use superscript for exponents.

Author: Done

Line 274. Add “the” before extrapolation.

Author: Done

Line 275. Change this sentence to “A Summary of extrapolation results with comparison to known complete counts is provided in S1 Table.”

Author: Changed

Line 282. Add “the” before walrus

Author: Added

Line 285. Not sure what more than one level means. More than one parameter?

Author: Suggested correction: We refrained from incorporating additional

---

## [Editor Report · Decision Letter 1]

2 Jul 2024

Enhancing the accuracy and efficiency of Pacific walrus (Odobenus rosmarus divergens) surveys: a comparison of visual and aerial imagery-based counts at coastal haulouts.

PONE-D-24-08990R1

Dear Dr. Altukhov,

Thank you for addressing the concerns and editing suggestions provided by the two reviewers of your manuscript. I am pleased to let you know that I find that your manuscript scientifically suitable for publication and I am recommending that it be formally accepted for publication once it meets any outstanding technical requirements identified by the Editorial Office.

Within one week, you’ll receive an e-mail detailing any required amendments. When these have been addressed, you’ll receive a formal acceptance letter and your manuscript will be scheduled for publication.

Thank you for your interest in having your important work on enumerating and ultimately protecting Pacific walrus populations in Chukotka and elsewhere in the Arctic published in PLOS One.

Kind regards,

Lee W Cooper, Ph.D.

Section Editor

PLOS ONE

---

## [Editor Report · Acceptance letter]

5 Jul 2024

PONE-D-24-08990R1 

PLOS ONE

Dear Dr. Altukhov, 

I'm pleased to inform you that your manuscript has been deemed suitable for publication in PLOS ONE. Congratulations! Your manuscript is now being handed over to our production team.

Kind regards, 

on behalf of

Dr. Lee W Cooper 

Section Editor

PLOS ONE